# Delta neutrophil index and shock index can stratify risk for the requirement for massive transfusion in patients with primary postpartum hemorrhage in the emergency department

**Taeyoung Kong[1], Hye Sun Lee[2], So Young Jeon[2], Je Sung You[1]***, **Jong Wook Lee[3], Hyun Soo Chung[1], Sung Phil Chung[1]**

1 Department of Emergency Medicine, Yonsei University College of Medicine, Seoul, Republic of Korea, 2 Department of Research Affairs, Biostatistics Collaboration Unit, Yonsei University College of Medicine, Seoul, Republic of Korea, 3 Department of Laboratory Medicine, Konyang University Hospital, Daejon, Republic of Korea

* youjsmd@yuhs.ac

## Abstract

### Background

Postpartum hemorrhage (PPH) constitutes a major risk for maternal mortality and morbidity. Unfortunately, the severity of PPH can be underestimated because it is difficult to accurately measure blood loss by visual estimation. The delta neutrophil index (DNI), which reflects circulating immature granulocytes, is automatically calculated in hematological analyzers. We evaluated the significance of the DNI in predicting hemorrhage severity based on the requirement for massive transfusion (MT) in patients with PPH.

### Methods

We retrospectively analyzed data from a prospective registry to evaluate the association between the DNI and MT. Moreover, we assessed the predictive ability of the combination of DNI and shock index (SI) for the requirement for MT. MT was defined as a transfusion of ≥10 units of red blood cells within 24 h of PPH. In total, 278 patients were enrolled in this study and 60 required MT.

### Results

Multivariable logistic regression revealed that the DNI and SI were independent predictors of MT. The optimal cut-off values of ≥3.3% and ≥1.0 for the DNI and SI, respectively, were significantly associated with an increased risk of MT (DNI: positive likelihood ratio [PLR] 3.54, 95% confidence interval [CI] 2.5–5.1 and negative likelihood ratio [NLR] 0.48, 95% CI 0.4–0.7; SI: PLR 3.21, 95% CI 2.4–4.2 and NLR 0.31, 95% CI 0.19–0.49). The optimal cut-off point for predicted probability was calculated for combining the DNI value and SI value with the equation derived from logistic regression analysis. Compared with DNI or SI alone,

**Data Availability Statement:** All relevant data are within the manuscript and its Supporting Information files.

**Funding:** This study was supported by the Basic Science Research Program of the National Research Foundation of Korea (NRF), funded by the Ministry of Science and ICT, in the form of a grant awarded to JSY (NRF- 2018R1C1B6006159), Yonsei University College of Medicine in the form of a faculty research grant for 2019 awarded to JSY (6-2019-0188), and Siemens Health Care in the form of research funding awarded to JSY (3-2015-0140), which did not exceed $10,000 per year. The funders had no role in study design, data collection and analysis, decision to publish, or preparation of the manuscript.

**Competing interests:** The authors have read the journal's policy and have the following competing interests: Siemens Health Care provided support for this study in the form of research funding awarded to JSY, which did not exceed $10,000 per year. The automated blood cell analyzer (ADVIA 2120; Siemens, Forchheim, Germany) is a marketed product associated with this research. This does not alter our adherence to PLOS ONE policies on sharing data and materials. There are no other patents, products in development or marketed products associated with this research to declare.

**Abbreviations:** PPH, Postpartum hemorrhage; DNI, delta neutrophil index; SI, shock index; MT, massive transfusion; ORs, odds ratios; HR, heart rate; SBP, systolic blood pressure; ED, emergency department; CBC, complete blood count; SPEED, postpartum blEeding through Expeditious care Delivery; CP, critical pathway; CPOE, computerized physician order entry; WBC, white blood cell; ICU, intensive care unit; ROC, receiver operating characteristic; AUCs, areas under the curve; Cis, confidential intervals; GEE, generalized estimating equations.

the combination of DNI and SI significantly improved the specificity, accuracy, and positive likelihood ratio of the MT risk.

## Conclusion

The DNI and SI can be routinely and easily measured in the ED without additional costs or time and can therefore, be considered suitable parameters for the early risk stratification of patients with primary PPH.

## Introduction

Postpartum hemorrhage (PPH) occurs in approximately 3–5% of obstetric patients [1]. Although it is uncommon, in approximately 20% of cases, PPH develops in patients without risk factors of bleeding [1]. PPH constitutes a major risk for maternal mortality and morbidity in both developed and developing countries and causes 12% of maternal mortality in the United States [1,2]. Generally, patients with a healthy pregnancy can tolerate blood loss of 500–1,000 mL with no symptoms or signs of hypovolemia [1,3]. PPH is defined as a blood loss of $\geq$ 500 mL, and severe PPH is defined as a blood loss of $\geq$ 1,000 mL. According to the American College of Obstetricians and Gynecologists, early PPH is defined as a total blood loss of or blood loss with symptoms and signs of hypovolemia of at least 1,000 mL within 24 h after intrapartum loss or delivery of fetus [1,4]. Primary PPH may develop before placenta delivery and up to 24 h after fetus delivery [1]. Prompt diagnosis and immediate resuscitation using a multidisciplinary team are required for the proper management of PPH [1,2]. Unfortunately, its severity can be underestimated because it is difficult to accurately measure blood loss by visual estimation, and there are no generally accepted cut-off limits for estimated blood loss [5]. Inaccurate estimation of blood loss can potentially lead to misleading clinical decisions [6,7]. Thus, objective parameters are needed for predicting hemorrhage severity and the requirement for massive transfusion in PPH patients [7]. The decrease in hematocrit is poorly correlated with blood loss severity and is clinically unavailable in emergency situations [5].

As traditional vital signs alone do not provide a reliable measurement of blood loss severity in the early stage after acute blood loss, some studies have demonstrated that the initial shock index (SI), defined as the ratio of the heart rate (HR) to the systolic blood pressure (SBP), was an independent predictor of the requirement for massive transfusion in patients with PPH [8,9]. An SI of >1.0 showed a positive likelihood ratio (PLR) of 2.8 and a negative likelihood ratio (NLR) of 0.5 for the requirement for massive transfusion in patients with PPH, indicating that the SI is of little help to clinicians for predicting the need for transfusion [7]. In general, likelihood ratios are only useful if their values are >5 or <0.2 [10]. Therefore, additional biomarkers or indicators are needed to better predict the requirement for massive transfusion in patients with PPH in the emergency department (ED). The DNI, which reflects the proportion of circulating immature granulocytes, is automatically calculated as the difference between the leukocyte differentials measured in the myeloperoxidase channel and those detected in the lobularity channel [11,12]. Many recent studies have founded that the DNI was significantly associated with the severity of diseases related to sterile inflammation, such as out-of-hospital cardiac arrest, major traumatic injury, pulmonary embolism, acute myocardial infarction, and upper gastrointestinal hemorrhage [12–16]. As an inflammatory response to hypoperfusion and reperfusion owing to blood loss and tissue damage can develop into systemic inflammatory response syndrome, the DNI is a promising candidate for a biomarker for predicting the requirement of massive transfusion in patients with primary PPH in the ED [12,13].

To the best of our knowledge, the usefulness of the DNI to stratify the severity of blood loss in patients with primary PPH in emergency settings has not been investigated. The aims of the present study were as follows: 1) to evaluate the clinical utility of the DNI as an indicator of disease severity based on the requirement for massive transfusion in patients with primary PPH admitted to the ED and 2) to investigate whether the combination of the initial DNI and SI, the latter being a hemodynamic indicator that can be easily obtained in the ED, can show improved predictive ability for the requirement for massive transfusion.

## Methods

### Study design and patients

This observational cohort study was retrospectively conducted based on a prospective registry of patients who presented with PPH in the ED of Severance Hospital between January 2011 and December 2019. The hospital is affiliated with Yonsei University College of Medicine and is a 2,500-bed tertiary-level referral hospital that serves approximately 95,000 patients per year. The study was approved by the institutional review board (IRB) of Yonsei University Health System (Seoul, Republic of Korea) (approval no.: 3-2017-0302), and the requirement for written informed consent was waived because of the retrospective nature of the study.

Since January 2009, the Severance Protocol to save postpartum blEeding through Expeditious care Delivery (SPEED) has been implemented as a critical pathway (CP) program based on computerized physician order entry (CPOE) at our institution as a part of a quality improvement initiative [2]. To provide prompt treatment and rapid transfusion to patients with primary PPH, we designed a standard protocol with an integrated alarm system using CPOE to reduce unnecessary delays in the hospital by implementing rapid decision-making by specialists. Primary PPH was defined as a hemorrhage requiring fluid resuscitation or transfusion within the first 24 h of delivery [7]. When a patient with PPH was approved for transfer from another hospital or a private clinic to our ED, emergency physicians and obstetricians were standing by in the ED to prepare for treatment, and the blood bank prepared universal O-type blood for transfusion. According to the predetermined protocol, an anesthesiologist and radiologist were also notified to prepare for emergency surgery or radiologic intervention using the CPOE and an automatic short message server [2]. Our institution schedules specialized obstetrics and interventional radiologists to be on call 24 h/day, 7 d/week, to treat patients who arrive at the ED requiring emergency surgical and radiological interventions [2]. Data on the patients' clinical characteristics, laboratory results, severity of the clinical condition, and treatment process were prospectively obtained from the SPEED CP registry using a predetermined protocol.

This study analyzed consecutive patients with primary PPH who were prospectively integrated into the SPEED CP program from among those admitted between January 1, 2011 and December 31, 2019. Hence, patients who were referred to our ED for the evaluation and management of PPH from other hospitals or private obstetric clinics within the first 24 h of delivery were also included in this study. We excluded patients who met any of the following exclusion criteria: cardiac arrest before ED admission, refusal to receive a transfusion, and a diagnosis of hemolysis, elevated liver enzymes, and low platelets syndrome. Fig 1 shows the enrollment, exclusion, and clinical outcome data of patients with PPH.

### Data collection

After obtaining the IRB approval, we accessed the SPEED registry database and collected data in January 2020. Data on the baseline and clinical characteristics of all patients; including age, parity, type of delivery, initial mental status, initial vital signs, initial laboratory findings, and

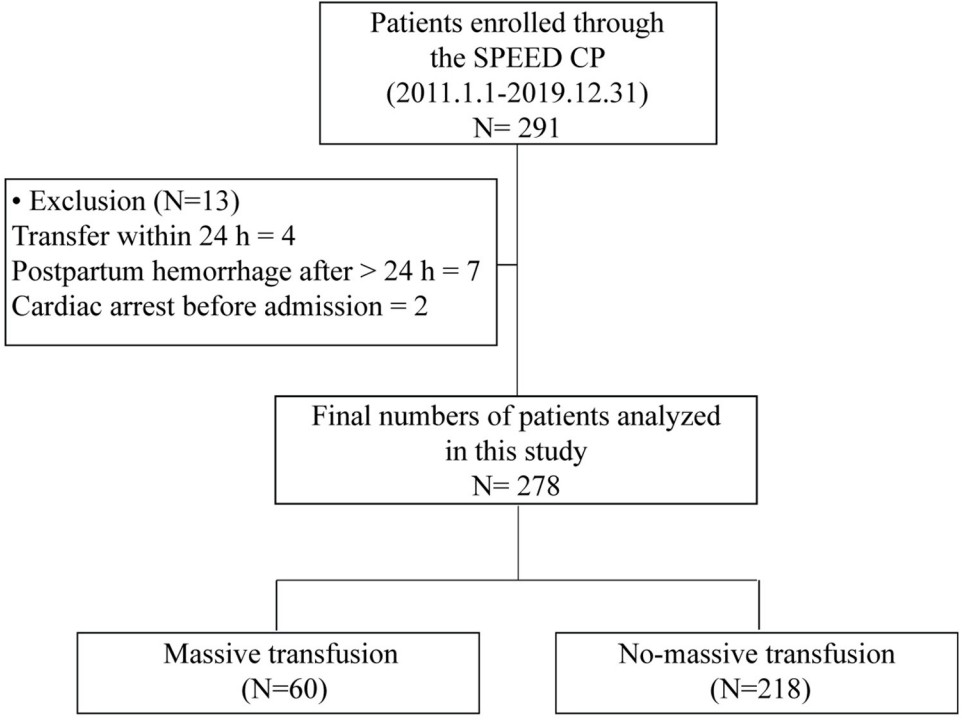

**Fig 1. Flow diagram of the enrollment of patients.**

amount of blood transfusion; and the clinical outcomes; including emergency surgery, embolization, hysterectomy, length of hospital stay, and in-hospital death; were retrieved from the prospective SPEED registry. Mental status was assessed at triage on ED admission using the Alert/Verbal/Painful/Unresponsive scale. Vital signs, including HR, body temperature, SBP, and diastolic BP, were also assessed at triage on ED admission. During the study period, emergency physicians or obstetricians treating the patient decided on the necessity, type, and amount of blood transfusion because there were no transfusion guidelines for patients with primary PPH and massive transfusion in our institution. Venous blood was collected in ethylenediaminetetraacetic acid-containing vacutainers within 10 min of ED admission for the measurement of the DNI using an automated blood cell analyzer (ADVIA 2120; Siemens, Forchheim, Germany) used for CBC analysis. We also performed other laboratory tests, including tests for prothrombin time, total $CO_2$, blood urea nitrogen, creatinine levels, albumin levels, and total bilirubin levels, at the time of ED admission.

## Patients' characteristics

In total, 278 patients with primary PPH were enrolled in this study. Of those, 60 patients (21.6%) required massive transfusion (Fig 1). Table 1 shows the baseline and clinical characteristics of patients classified according to the requirement for massive transfusion. There was no difference in parity between the two groups. However, there were significant differences in age, body temperature, mental status on ED admission, and type of delivery between the two groups. Patients who required a massive transfusion had a significantly lower SBP (94 ± 26 vs. 115 ± 20 mmHg, $p<0.001$) and diastolic BP (59 ± 18 vs. 69 ± 13 mmHg, $p<0.001$) and a higher HR (114 ± 23 vs 96 ± 20 bpm, $p<0.001$) on ED admission than those who did not require a massive transfusion. The initial SI on ED admission was significantly higher in patients who

**Table 1. Comparison of clinical characteristics and clinical outcomes according to the requirement for massive transfusion in patients with primary postpartum hemorrhage.**

| Variables | Total | Non-MT | MT | P |
|---|---|---|---|---|
| | N = 278 | N = 218 (78.4%) | N = 60 (21.6%) | |
| Age (years) | 32.70±3.95 | 32.37±3.93 | 34.03±3.78 | 0.004* |
| **Parity [n (%)]** | | | | 0.46 |
| Primipara | 169(60.79) | 135(61.93) | 34(56.67) | |
| Multipara | 109(39.21) | 83(38.07) | 26(43.33) | |
| **Type of delivery [n (%)]** | | | | 0.026* |
| Vaginal delivery | 182(65.47) | 150(68.81) | 32(53.33) | |
| Caesarean section | 96(34.53) | 68(31.19) | 28(46.67) | |
| **Initial mental status [n (%)]** | | | | <0.001* |
| Alert | 271(97.48) | 217(99.54) | 54(90.00) | |
| Non-alert | 7(2.52) | 1(0.46) | 6(10.00) | |
| **Initial vital sign** | | | | |
| Shock index | 0.95±0.35 | 0.86±0.24 | 1.30±0.45 | <0.001* |
| Shock index > 1 [n (%)] | 94 (33.81) | 48 (22.02) | 46 (76.67) | <0.001* |
| SBP (mmHg) | 111±23 | 115±20 | 94±26 | <0.001* |
| DBP (mmHg) | 67±15 | 69±13 | 59±18 | <0.001* |
| Heart rate (bpm) | 100±22 | 96±20 | 114±23 | <0.001* |
| Body temperature (°C) | 36.8±0.7 | 36.9±0.7 | 36.5±0.8 | 0.002* |
| **Initial laboratory data** | | | | |
| WBC count ($10^3/\mu L$) | 16.94±6.72 | 16.94±6.68 | 16.92±6.94 | 0.984 |
| Hemoglobin (g/dL) | 9.67±2.03 | 9.91±1.94 | 8.81±2.13 | <0.001* |
| Platelet count ($10^3/\mu L$) | 180.54±84.23 | 194.94±84.03 | 128.23±61.56 | <0.001* |
| Albumin (g/dL) | 2.37±0.62 | 2.50±0.57 | 1.98±0.60 | <0.001* |
| Total CO2 (mmol/L) | 17.74±3.08 | 18.17±2.80 | 16.19±3.52 | <0.001* |
| Arterial lactate (mmol/L) | 2.71±2.39 | 1.94±0.96 | 3.80±3.24 | <0.001* |
| Delta neutrophil index | 2.58±3.09 | 1.86±1.76 | 5.21±4.95 | <0.001* |
| **Blood transfusion** | | | | |
| Transfusion before ED arrival [n (%)] | 130(46.76) | 104(43.33) | 26(68.42) | 0.004* |
| pRBCs before ED arrival (unit) | 1.66±2.90 | 1.27±2.10 | 4.11±5.23 | 0.002* |
| Total pRBCs (unit) | 6.32±7.06 | 3.39±2.46 | 16.98±8.04 | <0.001* |
| Total fresh frozen plasma (unit) | 2.04±3.81 | 0.68±1.42 | 6.95±5.42 | <0.001* |
| Total platelet concentrate (unit) | 2.49±6.52 | 0.62±2.21 | 9.27±11.05 | <0.001* |
| **Clinical outcome [n (%)]** | | | | |
| Embolization | 199(71.58) | 151(69.27) | 48(80.00) | 0.103 |
| Hysterectomy | 15(5.40) | 1(0.46) | 14(23.33) | <0.001* |
| Emergent surgery | 42(15.11) | 9(4.13) | 33(55.00) | <0.001* |
| In-hospital death | 1(0.36) | 0(0.00) | 1(1.67) | 0.216 |

*$P<0.05$. MT, massive transfusion; SBP, systolic blood pressure; DBP, diastolic blood pressure; WBC, white blood cell; ED, emergency department; pRBCs, packed red blood cells.

required a massive transfusion than in those who did not require a massive transfusion (1.30 ± 0.48 vs 0.86 ± 0.24, $p<0.001$). Patients who received a massive transfusion had a higher DNI than those who did not (5.21 ± 4.95 vs 1.86 ± 1.76%, $p<0.001$). In addition, they had lower hemoglobin, platelet, total $CO_2$, and albumin levels. Considering therapeutic management, patients who received a massive transfusion required more units of blood components and required emergency surgery and hysterectomy more often than those who did not receive

a massive transfusion. In addition, they had a longer ICU admission ($1.7 \pm 1.6$ vs. $0.3 \pm 0.7$ days, $p<0.001$). However, in-hospital mortality was not significantly different between the two groups; only one case of death was observed in our study, which occurred in a patient who required a massive transfusion.

## DNI measurement

The automated blood cell analyzer could determine the CBC count, comprising the DNI, white blood cell (WBC) count, hemoglobin level, and platelet count [12,17]. The specific analyzers used to determine the DNI were optical systems based on a cytochemical myeloperoxidase tungsten-halogen channel (which evaluates and differentiates eosinophils, lymphocytes, monocytes, neutrophils, and large unstained cells based on myeloperoxidase staining intensity and cell size) and a laser-diode channel (which calculates, counts, and classifies cell types based on lobularity/nuclear density and size) [12,17–19]. The DNI was then calculated by subtracting the fraction of mature polymorphonuclear neutrophils from the sum of myeloperoxidase-reactive cells, detecting circulating immature granulocytes as a leukocyte sub-fraction [12,17,18]. The DNI was obtained using the following formula: DNI = (neutrophil sub-fraction + eosinophil sub-fraction measured in the myeloperoxidase channel)—(polymorphonuclear sub-fraction measured in the nuclear lobularity channel) [11–13].

## SI measurement

The following formula was used for calculating the SI: SI = HR/SBP. The SI on ED admission was calculated based on the vital signs on ED admission.

## Clinical endpoints

The primary endpoint was the requirement for massive transfusion within 24 h of the development of PPH. Massive transfusion was defined as a transfusion of $\geq$ 10 units of packed red blood cells. To determine the amount of transfusion, we measured both the amount of transfused blood before arrival at the ED and the amount of transfused blood after arrival at the ED (in the ED and in the intensive care unit [ICU] or general ward).

## Statistical analysis

The data are presented as mean (SD) for continuous variables and as absolute or relative frequencies for categorical variables. Continuous variables were compared using the two-sample t-test or Mann-Whitney U-test, and categorical variables were compared using the $\chi 2$ test or Fisher's exact test. Univariate logistic regression analysis was performed to identify relationships between demographic characteristics and clinical data. A multivariate logistic regression analysis integrated the major covariates identified in the univariate analyses (variables with a $p<0.05$). A multivariate logistic regression analysis was also performed to assess promising independent factors for predicting the requirement for massive transfusion. We plotted receiver operating characteristic (ROC) curves and determined the areas under the curve (AUCs) to evaluate the ability of the DNI and SI to predict the requirement for massive transfusion. Youden's method was used to verify the optimal cut-off of the DNI and SI for discriminating between the requirement and non-requirement for massive transfusion. These results are presented as odds ratios (ORs) and 95% confidential intervals (CIs). In addition, we computed the ROC curve to determine the predictive accuracy and identified the sensitivity, specificity, accuracy, PLR, and NLR for predicting the requirement for massive transfusion considering the DNI, SI, and the combination of the DNI and SI; we also calculated the optimal cut-off values. In addition, to stratify risk for the

requirement for massive transfusion by combining the DNI value and SI value, we used the equation derived from logistic regression analysis. Based on this equation, we calculated the predicted probability for combining the DNI value and SI value. We determined optimal the cut-off point for the predicted probability. Based on this optimal cut-off point, we measured sensitivity, specificity, PLR, NLR, and accuracy. The generalized estimating equation (GEE) method was used to compare the significance of the DNI, SI, and the combination of the DNI and SI. Statistical analyses were performed using SAS version 9.2 (SAS Institute Inc., Cary, NC, USA) and MedCalc Statistical Software version 16.4.3 (MedCalc Software bvba, Ostend, Belgium), and a *p* value < 0.05 was considered significant.

## Results

### DNI and SI as predictors of the requirement for massive transfusion

Multivariate logistic regression analysis was used to identify independent factors associated with the requirement for massive transfusion (Table 2). Based on the univariate analysis results, initial mental status, hemoglobin level, serum total $CO_2$ level, albumin concentration, SI, and DNI were included in the multivariate logistic regression analysis (S1 Table). The requirement for massive transfusion was not independently associated with the hemoglobin and serum total $CO_2$ levels on ED admission. The DNI value was independently associated with the requirement for massive transfusion (OR: 1.247, 95% CI: 1.072–1.451, *p* = 0.004).

The SI was also independently associated with the requirement for massive transfusion (OR: 13.73, 95% CI: 3.46–54.46; *p*<0.001). In the ROC curve analysis of the requirement for massive transfusion, the AUC values of the DNI and SI were 0.74 (95% CI: 0.66–0.82; *p*<0.001) and 0.82 (95% CI: 0.76–0.89; *p*<0.001), respectively (Fig 2). Using Youden's method, the optimal cut-off values for the DNI and SI on ED admission were found to be 3.3% and 1, respectively.

An increase in the DNI of ≥3.3% was significantly associated with the requirement for massive transfusion, with a specificity of 83% (95% CI: 78–88), sensitivity of 60% (95% CI: 47–72), accuracy of 78% (95% CI: 73–83), PLR of 3.54 (95% CI: 2.47–5.06), and NLR of 0.48 (95% CI; 0.35–0.66) (Table 3). An increase in the SI of ≥ 1.0 was also significantly associated with the requirement for massive transfusion, with a specificity of 76% (95% CI: 70–82), sensitivity of 77% (95% CI: 66–87), accuracy of 76% (95% CI: 71–81), PLR of 3.21 (95% CI: 2.44–4.23), and NLR 0.31 (95% CI: 0.19–0.49).

### Prognostic value of the combination of the DNI and SI for the requirement for massive transfusion

Considering the results for the SI and DNI on ED admission, we found that the SI had a lower specificity and the DNI had a lower sensitivity and a higher specificity for predicting the

**Table 2. Factors associated with the requirement for massive transfusion in patients with primary postpartum hemorrhage in multivariate analysis.**

| Variable | Unadjusted OR (95% CI) | P | Adjusted OR† (95% CI) | P |
|---|---|---|---|---|
| Hemoglobin (per 1g/dL) | 0.759 (0.653–0.881) | <0.001* | 0.911 (0.745–1.114) | 0.365 |
| Total CO2 (per 1mmol/L) | 0.802 (0.723–0.888) | <0.001* | 0.990 (0.865–1.132) | 0.881 |
| Albumin (per 1g/dL) | 0.181 (0.097–0.339) | <0.001* | 0.372 (0.172–0.805) | 0.012* |
| Mental change (vs alert) | 24.108 (2.843–204.450) | 0.004* | 14.744 (1.319–164.838) | 0.029* |
| Delta neutrophil index | 1.490 (1.305–1.701) | <0.001* | 1.247 (1.072–1.451) | 0.004* |
| Shock index | 56.292 (17.269–183.498) | <0.001* | 13.730 (3.462–54.458) | <0.001* |

*P<0.05. OR, odds ratio; CI, confidence interval

†Multivariate model including all the variables listed in the table.

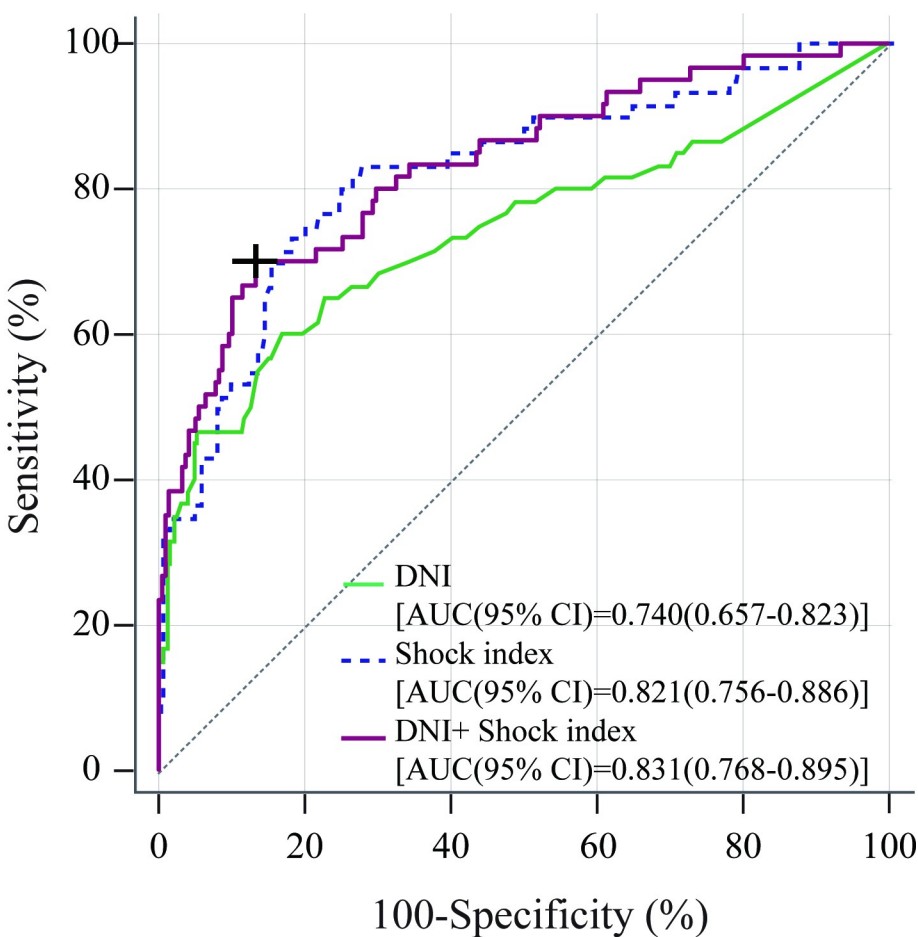

**Fig 2. Receiver operating characteristic curves of the delta neutrophil index, shock index, and the combination of the delta neutrophil index and shock index for the requirement for massive transfusion in patients with primary postpartum hemorrhage.** DNI, delta neutrophil index; SI, shock index; AUC, area under the curve; CI, confidence interval.

requirement for massive transfusion. To compensate for the weaknesses of both indicators, we measured the sensitivity, specificity, accuracy, PLR, and NLR of the combination of the DNI and SI. On combining a DNI value of ≥3.3 with an SI value of ≥1.0, the specificity and PLR increased to 95% (95% CI; 93–98) and 11.63 (95% CI: 6.07–22.27), respectively. However, the sensitivity and NLR were 53% (95% CI: 41–66) and 0.49 (95% CI: 0.37–0.64), respectively.

The predicted probability derived from the logistic equation was computed for each patient. When the predicted probability of the equation [Predicted probability = $1/(1+\exp(5.348-0.2464^*DNI-3.2691.^*SI)]$ containing the combination of the DNI and SI was more than 0.249, the specificity was 87% (95% CI: 82–91), sensitivity was 70% (95% CI: 58–82), accuracy was 83% (95% CI: 79–88), PLR was 5.26 (95% CI: 3.6–7.7), and NLR was 0.35 (95% CI: 0.2–0.5) for MT. The combination of the DNI and SI significantly improved the specificity (DNI vs DNI + S, $p<0.001$; SI ≥1.1 vs DNI + SI, $p<0.001$), accuracy (DNI vs DNI + SI, $p<0.001$; SI ≥1.1 vs DNI + SI, $p<0.001$), and PLR (DNI vs DNI + S, $p<0.001$; SI ≥1.1 vs DNI + SI, p = 0.038; using the GEE method) for the requirement for massive transfusion when compared with either DNI or SI (Table 3).

**Table 3. Prognostic value of the shock index, delta neutrophil index, and combined delta neutrophil index with shock index for predicting the requirement for a massive transfusion in patients with primary postpartum hemorrhage.**

| Variable | Sensitivity (95% CI) | Specificity (95% CI) | Accuracy (95% CI) | PLR | NLR |
|---|---|---|---|---|---|
| | | | | (95% CI) | (95% CI) |
| 1> Delta neutrophil index ≥ 3.3 | 60 | 83 | 78.1 | 3.54 | 0.48 |
| | (47.6–72.4) | (78.1–88.0) | (73.2–82.9) | (2.47–5.06)) | (0.35–0.66) |
| 2> Shock index ≥ 1 | 76.7 | 76.2 | 76.3 | 3.21 | 0.31 |
| | (66.0–87.4) | (70.5–81.8) | (71.3–81.3) | (2.44–4.23) | (0.19–0.49) |
| 3> Predicted probability§ ≥ 0.249 | 70 | 86.7 | 83.1 | 5.26 | 0.35 |
| | (58.4–81.6) | (82.2–91.2) | (78.7–87.5) | (3.61–7.67) | (0.23–0.51) |
| | Comparison of P value | | | | |
| 1 vs 3 | 0.038* | <0.001* | <0.001* | <0.001* | 0.853 |
| 2 vs 3 | <0.001* | <0.001* | <0.001* | <0.001* | 0.015* |

*P<0.05. PLR, positive likelihood ratio; NLR, negative likelihood ratio; CI, confidence interval.

§Equation of predicted probability = 1/(1+exp (5.348–0.2464*DNI-3.2691*SI).

## Discussion

We demonstrated that the DNI on ED admission, which reflects the fraction of immature granulocytes, was independently associated with the requirement for massive transfusion in patients with primary PPH, with an AUC value of 0.74. Using the optimal cut-off value of 3.3% for the DNI on ED admission, the DNI was significantly associated with an increased risk of massive transfusion, with a PLR of 3.54 and an NLR of 0.48. To the best of our knowledge, this study is the first to identify the usefulness of the DNI on ED admission for predicting the requirement for massive transfusion in patients with primary PPH.

The optimal cut-off value for the DNI on ED admission was 3.3% in this study. Yune et al. found that a DNI >10.5% on day 1 significantly predicted a higher 30-day mortality rate after out-of-hospital cardiac arrest [16]. In a study on trauma patients admitted to an ICU, Kong et al. proposed that DNI values of >3.25% and >5.3% at 12 h post-admission were significantly associated with the development of multiple organ dysfunction syndrome and 30-day mortality, respectively [12]. Kong et al. also demonstrated that the optimal cut-off value of the DNI was 1% at admission (p<0.001) and 2.6% on day 1 (p<0.001) for predicting 30-day mortality in patients with upper gastrointestinal bleeding [13]. Taken together, these studies suggested that the DNI values over time reflect the severity of diseases related to systemic and sterile inflammation without infection [12].

The DNI is available in emergency settings because it can be rapidly and easily determined using an automated blood cell analyzer while measuring the CBC [11–13] The DNI value is correlated with the manual count of immature granulocytes [11]. However, automatic calculation of the DNI is a more prompt and accurate method than manual counting of immature granulocytes [12]. Several studies have shown that an increase in the DNI can predict the severity of infection and sterile inflammation without infection and that it is significantly associated with poor outcomes in specific disease conditions [12–14,17,18,20,21]. Emergency granulopoiesis is the most accepted theory for explaining why an elevated DNI value is associated with the use of massive transfusion after primary PPH [22,23]. Emergency granulopoiesis implies the expansion of the immature granulocytic population for providing hematopoietic precursors to sustain the increasing number of circulating neutrophils [12,22,23]. The number of circulating immature neutrophils can be rapidly increased to compensate for the destruction of mature neutrophils due to severe systemic inflammation and the loss of active

neutrophils due to massive consumption related to sterile stimuli, such as chemical agents, trauma, bleeding, and ischemic insult [22,24,25]. Livingston et al. proposed that bone marrow failure can develop due to insufficient perfusion and ischemia during hemorrhagic shock [26,27]. This bone marrow dysfunction causes a transient failure of granulopoiesis [26,27]. Consequently, severe hemorrhage may contribute to the release of immature granulocytes into the circulation [24,28]. As pregnant women often have prenatal anemia, it is difficult to determine the occurrence of massive hemorrhage and evaluate the requirement for massive transfusion based on hemoglobin levels alone in patients with PPH [29]. Although lower hemoglobin or hematocrit levels have been widely and interchangeably used as indicators of severe bleeding, their diagnostic values for determining the severity of blood loss in the initial phase in patients with severe hemorrhage, such as trauma-related and GI bleeding, remain controversial [29]. In the present study, the hemoglobin level on ED admission was not associated with the requirement for massive transfusion.

In addition, some studies have reported that the initial SI value, as a hemodynamic indicator based on the HR and SBP (easily obtained in the ED), could predict the requirement for massive transfusion. Although the initial SI value on ED admission was independently associated with the requirement for massive transfusion in patients with primary PPH, it had a relatively low specificity [7,8].

In the present study, we found that using a combination of the DNI and SI significantly improved the specificity, accuracy, and PLR for the requirement for massive transfusion in patients with primary PPH when compared with either DNI or SI.

As PPH is significantly associated with a high risk of maternal death, it is crucial to assess and hemodynamically stabilize patients in a timely manner by careful monitoring [2]. In high-risk patients, prompt resuscitation and treatment should be initiated to decrease maternal mortality and morbidity [2]. Compared with the findings of Sohn et al., in our study, the SI had a relatively high sensitivity. This suggested its clinical implication: the initial SI value can be used as a screening tool for the initial risk stratification on ED admission, while the combination of the SI and DNI can be used as an additional prediction tool for the requirement for massive transfusion in patients with primary PPH.

This study had several limitations. First, although the data were obtained from the prospectively maintained SPEED CP registry using a standardized and predetermined protocol at our institution, they were obtained from a single, tertiary, academic hospital and were analyzed retrospectively. Furthermore, as the present study included patients with PPH who were transferred from other hospitals or private clinics to our ED, we had to rely on the medical records of the referring hospital or clinic to identify the amount of transfused blood before ED admission at our institution. Therefore, it was difficult to completely control for confounding factors, which increased the potential risk of selection bias. Second, we could not obtain accurate data related to the time of development of massive bleeding in the referring hospitals and clinics. For minimizing the effect of prehospital transfusions, we included patients with ED admission within 24 h of the onset of primary PPH. Third, a previous study demonstrated that the DNI value was an independent predictor of the development of multiple organ dysfunction syndrome and short-term mortality in trauma patients who were admitted to an ICU, but its predictive ability was not affected by the amount of transfusion [12]. As transfused blood products may contain WBCs, prehospital transfusions may have affected the DNI value. However, the WBCs in the blood products may get diluted in the blood of the whole body; thus, the effect on the DNI value may be trivial. Nevertheless, further prospective, randomized, multicenter trials are required to assess the prehospital time factor, the effect of prehospital transfusions, and the clinical applicability of our results.

## Conclusions

We demonstrated that an increased DNI is an independent predictor of the requirement for massive transfusion in patients with primary PPH admitted to the ED. Combining the initial DNI with the initial SI can improve the predictive ability for the requirement for massive transfusion. The DNI and SI can be routinely and easily measured in the ED without additional costs or time and can therefore, be considered suitable parameters for the early risk stratification of patients with primary PPH. In terms of clinical implications, the initial SI value can be useful as a screening tool for risk stratification on ED admission, and the combination of the SI and DNI can be applied as a confirmative prediction tool for the requirement for massive transfusion in patients with primary PPH.

## Supporting information

**S1 Table. Univariate logistic regression analysis for clinical factors associated with the requirement for massive transfusion in patients with primary postpartum hemorrhage.** (DOCX)

**S2 Table. Original data that was used in the study.** (XLSX)

## Author Contributions

**Conceptualization:** Taeyoung Kong, Je Sung You, Jong Wook Lee.

**Data curation:** Taeyoung Kong, Je Sung You.

**Formal analysis:** Taeyoung Kong, Hye Sun Lee, So Young Jeon, Je Sung You.

**Funding acquisition:** Je Sung You.

**Investigation:** Je Sung You, Jong Wook Lee, Hyun Soo Chung, Sung Phil Chung.

**Methodology:** Taeyoung Kong, Hye Sun Lee, So Young Jeon, Je Sung You.

**Project administration:** Je Sung You.

**Supervision:** Je Sung You.

**Validation:** Je Sung You, Jong Wook Lee, Hyun Soo Chung, Sung Phil Chung.

**Visualization:** Taeyoung Kong.

**Writing – original draft:** Taeyoung Kong, Je Sung You.

**Writing – review & editing:** Taeyoung Kong, Hye Sun Lee, So Young Jeon, Je Sung You, Jong Wook Lee, Hyun Soo Chung, Sung Phil Chung.

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
