## [Decision Letter · Decision Letter 0]

26 Apr 2021

PONE-D-20-31096

Delta neutrophil index and shock index can predict the requirement for massive transfusion in patients with primary postpartum hemorrhage in the emergency department

PLOS ONE

Dear Dr. Je Sung You,

Thank you for submitting your manuscript to PLOS ONE. After careful consideration, we feel that it has merit but does not fully meet PLOS ONE’s publication criteria as it currently stands. Therefore, we invite you to submit a revised version of the manuscript that addresses the points raised during the review process.

ACADEMIC EDITOR: The reviewers have raised a number of points which we believe major modifications are necessary to improve the manuscript, taking into account the reviewers' remarks. Please consider and address each of the comments raised by the reviewers before resubmitting the manuscript. This letter should not be construed as implying acceptance, as a revised version will be subject to re-review.

We look forward to receiving your revised manuscript.

Kind regards,

Wisit Cheungpasitporn, MD

Academic Editor

PLOS ONE

Journal Requirements:

2)  Thank you for stating the following in the Competing Interests section:

[NO The authors have declared that no competing interests exist.]. 

We note that you received funding from a commercial source: Siemens Health Care

Reviewers' comments:

Reviewer's Responses to Questions

**Comments to the Author**

1. Is the manuscript technically sound, and do the data support the conclusions?

Reviewer #1: Partly

Reviewer #2: Partly

Reviewer #3: Yes

Reviewer #4: Partly

2. Has the statistical analysis been performed appropriately and rigorously? 

Reviewer #1: No

Reviewer #2: Yes

Reviewer #3: Yes

Reviewer #4: Yes

3. Have the authors made all data underlying the findings in their manuscript fully available?

Reviewer #1: No

Reviewer #2: Yes

Reviewer #3: Yes

Reviewer #4: Yes

4. Is the manuscript presented in an intelligible fashion and written in standard English?

Reviewer #1: Yes

Reviewer #2: Yes

Reviewer #3: Yes

Reviewer #4: Yes

5. Review Comments to the Author

Reviewer #1: 1. Abstract (background): The study purpose is not described.

2. Abstract (methods): “…to evaluate the usefulness of the DNI as an indicator of sterile inflammation severity” This is a vague description. Please indicate the purpose clearly.

3. Abstract (result: “Of the 278 patients enrolled, 60 required MT.” This sentence should be moved to methods section.

4. Keywords: “immature neutrophils” can be deleted.

5. Introduction: There are several definitions of PPH, including severe PPH, early PPH, and primary PPH. How can these definitions be applied to the study population? This information can be presented in Table 1 or Figure 1.

6. Introduction (page 6): Most of the paragraph is composed of lengthy, descriptive narration on DNI application in various clinical situations. This part can be shortened compactly.

7. Methods (page 7): How many patients were transferred from another hospital or a private clinic to ED? How many of them were transfused already before ED arrival? These baseline data should be presented in Table 1 or Figure 1.

8. Methods (page 9): During the 10-yr study period, was the same instrument (ADVIA 2120) used continuously without change?

9. Methods (page 10): In line with my comment #7, clinical endpoints “the amount of transfused blood before ED arrival and the amount of transfused blood after ED arrival” can be presented in Table 1 or Figure 1.

10. Statistical analysis: Although it is described that “The data are presented as medians and interquartile ranges for continuous variables”, the data are presented as mean/SD in Table 1.

11. Statistical analysis (page 11): diagnostic performance? diagnostic accuracy? Is it for the diagnosis of massive transfusion or for the prediction of massive transfusion? If it is for the prediction, throughout the text, predictive performance should be described instead of diagnostic performance.

12. The whole “RESULTS” section has lots of duplicated descriptions of each Table and Figure. Please avoid duplicated presentation. Moreover, the first subsection, “Patients’ characteristics”, should be moved to the Methods section. In this retrospective study, this part is not the result corresponding to the study purpose but the basic description on the study population.

13. Table 2 can be modified to show the univariate analysis results together. That would be much more comprehensive.

14. Fig. 2: Did the authors compare AUC values among DNI, SI, DNI + SI? Please provide the statistical data (pairwise comparison of AUC values). The comparison of 0.821 vs. 0.831 (with overlapping CIs) does not seem to provide a significant result.

15. Discussion section is quite lengthy spanning 4.5 pages (including Conclusion). It should be shortened compactly. The third paragraph (pages 14 – 16) is full of general descriptions that are not directly related to the study results. In the fourth paragraph (page 16), what was the lactate levels in this study population? If lactate levels are so important (as described in this paragraph), it deserves to be included in Table 1 and can be analyzed together.

16. This study used only the initial DNI values at ED admission. Follow-up DNI values would be available in these patients. Considering the time definition of massive transfusion, what was the DNI value and shock index value after 24 hrs?

17. In Table 1, how many patients had shock index > 1.0 in the two groups? The laboratory data should be presented in SI units.

18. In addition to the ROC curve analysis, IDI/NRI analyses are recommended to show additive value (any superiority) of DNI value to shock index.

Overall, the readability and organization of this manuscript is acceptable. However, the additive value of DNI value on top of shock index should be more emphasized to support their conclusion. Although the authors mentioned “screening tool” and “additional prediction tool” as the clinical usefulness of DNI value on top of shock index (page 17), which change in detail can we expect in the clinical practice? Can we decrease the transfused blood amount? Can we decrease the number of patients who were transfused massively? The authors performed a retrospective analysis; so, if they want to emphasize the adding value of DNI onto shock index, they can propose an algorithm (including both DNI value and shock index) for the clinical decision-making and can show the simulation results how the massive transfusion practice would have been changed.

Reviewer #2: The authors demonstrate a complex statistical association to integrate into daily medical practice. Moreover, the primary and secondary objectives are not at all clear. The method lacks precision; it is a study with a change of initial destination, but what was the purpose of the first study? what were the criteria?

The sample is small and few values are really significant.

This article does not provide any real benefit to the medical literature

Reviewer #3: Dear Editor,

Thanks for the opportunity to read the article.

It was a well-written and well-designed study that should be accepted for publication.

My recommendation is to accept the article in its current form.

Best regards.

Reviewer #4: The manuscript lines are not numbered, so I identified the lines of the manuscript by quoting parts of them.

Introduction

“…is the cause of maternal mortality in 12% of cases…” I assume what you mean is that 12% of maternal mortality is caused by PPH. Is that correct?

“An SI of >1.0 showed a specificity of 78.7%, a sensitivity of 59.2%, a positive predictive value

(PPV) of 58.7%, and a negative predictive value (NPV) of 79.1%.” Predictive values, although often reported, are useless statistics because they change with the prevalence of disease and you are trying to estimate the prevalence of disease in the individual patient. predictive values have almost no clinical usefulness, so please remove them from the paper. Although sensitivities and specificities are somewhat useful, likelihood ratios are more useful since they can be applied to the individual patient. Therefore, better would be a statement such as, “Using the results from reference 7, for a SI > 1.0, the positive and negative likelihood ratios for massive transfusion in patients with PPH are 3 and the negative likelihood ratio is 0.5, making SI of little help to the clinician for predicting the need for transfusion.” In general likelihood ratios are useful only if they are >5 or <0.2.

Methods

“…and identified…PPV, and NPV…” As mentioned above, PPV and NPV are useless statistics. Please eliminate them from the paper.

Results

“…SBP (94.267…). Reporting SBP to 3 significant figures is not needed. Better to report the results for blood pressures and heart rates as whole numbers. This will also make the paper easier to read.

I request two general changes in the paper. One has to do with the presentation of the results. As mentioned above, please remove predictive values and include likelihood ratios in your results. The second problem is the interpretation of the ROC curves as described next.

Fig. 2 The ROC curves for the DNI and DNI+SI are almost overlapping so it is slightly surprising that your sensitivities and specificities (Table 3) are so different. I think this is most likely because you have chosen different points on the curves to calculate them. I have described the method I use below, that I think is preferable. Nonetheless, the likelihood ratios calculated from the sensitivities and specificities listed in Table 3 are shown in the following table. Using these likelihood ratios, I have also calculated the post-test probabilities for need for massive blood transfusion, assuming that the pretest probability is 21.6% (60/278).

LR+ LR- Post-test probability if test positive Post-test probability if test is negative

DNI 3.5 0.48 49% 11%

SI 3.3 0.31 47% 8%

DNI + SI 11.6 0.49 76% 12%

The pretest probability of MT requirement is 21.6%. The question is, does applying DNI and/or SI change that significantly enough to benefit the clinician seeing the next patient that may require MT (and, if so, what does the clinician do differently than would not have been done if this information was not available)? As you can see, adding DNI to SI is not helpful if the test is negative, but somewhat helpful if the test is positive. In general, applying the results of your study to patients with PPH would reduce the probability of requiring MT in patients with a negative test by half, yet 10% of those patients still require MT. On the other hand, of those with a positive test, about one quarter will end up not requiring MT, using the DNI+SI parameters. Please comment on how this would help the clinician actually caring for these patients, in particular, once this result is obtained, how does the plan of the physician change? Would the clinician then immediately initiate the massive transfusion protocol, knowing that about one-quarter of the patients do not need it? I think that this type of analysis points out that even when differences are highly statistically significant, this does not necessary translate into a result that is that useful to the clinician seeing the next patient with PPH.

I performed two other calculations on your data. First, I had some difficulty with data taken from the DNI+SI ROC graph, since the point on the graph where sensitivity = 53.5% and specificity = 95.4% is not actually on the line. However, the point on the SI and SI+DNI lines closest to this point show the following results:

LR+ LR- Post-test probability if test positive Post-test probability if test is negative

SI 5.6 0.54 60% 13%

DNI + SI 7.3 0.53 67% 13%

This suggests that adding DNI to the SI is only marginally helpful. This is particularly important for two reasons: 1. The SI is immediately available once vital signs are taken and 2. Many clinicians may not be familiar with DNI and/or may have difficulty remembering the formula to calculate it or be able to calculate it themselves.

I actually calculated the same values by finding the points on the curves in Fig. 2 that are closest to the left upper corner of the graph (where sensitivity and specificity = 1). I did this by printing the graph and using a compass to find the point. This produced the following table, corresponding to the one above.

LR+ LR- Post-test probability if test positive Post-test probability if test is negative

DNI 2.8 0.45 43% 11%

SI 3.8 0.30 41% 7%

DNI+SI 5.0 0.35 57% 9%

As you can see, the results are similar to yours except for the DNI+SI with a positive result. This also suggests that adding DNI to SI is only a little helpful If the test is positive, but not at all if the test is negative.

6. PLOS authors have the option to publish the peer review history of their article (what does this mean?). If published, this will include your full peer review and any attached files.

Reviewer #1: No

Reviewer #2: No

Reviewer #3: No

Reviewer #4: **Yes: **Barnet Eskin

---

## [Author Response · Author response to Decision Letter 0]

17 Jul 2021

Reviewer #1: 

1. Abstract (background): The study purpose is not described. 

Response: Thank you for your comment. We agree with you. We have revised the abstract accordingly, as follows:

“Background: Postpartum hemorrhage (PPH) constitutes a major risk for maternal mortality and morbidity. Unfortunately, the severity of PPH can be underestimated because it is difficult to accurately measure blood loss by visual estimation. The delta neutrophil index (DNI), which reflects the concentration of circulating immature granulocytes, is automatically calculated in hematological analyzers. We evaluated the significance of the DNI in predicting hemorrhage severity based on the requirement for massive transfusion (MT) in patients with PPH.”

2. Abstract (methods): “…to evaluate the usefulness of the DNI as an indicator of sterile inflammation severity” This is a vague description. Please indicate the purpose clearly.

Response: Thank you for your comment. We agree with your comment. We have revised the abstract accordingly, as follows:

“Methods: We retrospectively analyzed data from a prospective registry to evaluate the association between the DNI and MT. Moreover, we assessed the predictive ability of the combination of DNI and shock index (SI) for the requirement for MT. MT was defined as a transfusion of ≥10 units of red blood cells within 24 h of PPH. In total, 278 patients were enrolled in this study and 60 required MT.”

3. Abstract (result: “Of the 278 patients enrolled, 60 required MT.” This sentence should be moved to methods section.

Response: Thank you for your comment. We agree with your suggestion. We have revised the abstract accordingly, as shown above in the response to Comment No. 2. 

4. Keywords: “immature neutrophils” can be deleted.

Response: Thank you for your comment. In line with your recommendation, we have removed the term “immature neutrophils” from the list of keywords.

5. Introduction: There are several definitions of PPH, including severe PPH, early PPH, and primary PPH. How can these definitions be applied to the study population? This information can be presented in Table 1 or Figure 1.

Response: Thank you for your comment. 

You have pointed out that patients enrolled in this study should be classified according to the three definitions of PPH: severe, early, and primary. In the Introduction section, we have presented the general characteristics and various definitions of PPH in relation to various related studies. 

PPH is defined as a blood loss of ≥ 500 mL, and severe PPH is defined as a blood loss of ≥ 1,000 mL. According to the American College of Obstetricians and Gynecologists, early PPH is defined as a total blood loss of at least 1,000 mL or blood loss with symptoms and signs of hypovolemia within 24 h after intrapartum loss or delivery of fetus [1,4]. Primary PPH may develop before placenta delivery and up to 24 h after fetus delivery [1]. In the present study, all enrolled patients had primary PPH and experienced bleeding within 24 h after fetus delivery. In addition, since all patients were transferred from another hospital or a private clinic due to severe bleeding, it can be considered that they all met the criteria for severe and early PPH.

To accurately classify PPH as severe or early based on the above definitions, it is necessary to estimate the amount of blood loss. However, as suggested in previous studies, it is difficult to accurately estimate the amount of blood loss in clinical settings. In our study, since all patients were transferred from another hospital or a private clinic due to severe bleeding, this would be very difficult. Moreover, we believe that classifying patients based on retrospectively estimated amount of blood loss may lead to erroneous interpretation of the study results.

The purpose of this study was to indirectly estimate the amount of blood loss in PPH, which is difficult to accurately measure, using the DNI and DNI+SI and predict the requirement for MT. Therefore, it is difficult to accurately classify patients based on these definitions of PPH.

6. Introduction (page 6): Most of the paragraph is composed of lengthy, descriptive narration on DNI application in various clinical situations. This part can be shortened compactly.

Response: Thank you for your comment. As we agreed with your comment, we have revised the Introduction section as follows:

An SI of >1.0 showed a positive likelihood ratio of 2.8 and a negative likelihood ratio of 0.5 for the requirement for massive transfusion in patients with PPH, indicating that the SI is of little help to clinicians for predicting the need for transfusion [7]. In general, likelihood ratios are only useful if their values are >5 or <0.2. 

However, additional biomarkers or indicators are needed to better predict the requirement for massive transfusion in patients with PPH in the emergency department (ED). With the advancement in technology, the delta neutrophil index (DNI) can be obtained using automated hematological analyzers based on differential leukocyte counts obtained from two independent channels[10,11]. The DNI, which reflects the proportion of circulating immature granulocytes, is automatically calculated as the difference between the leukocyte differentials measured in the myeloperoxidase channel and those detected in the lobularity channel[10,11]. Many recent studies have demonstrated that a higher DNI indicates higher severity in infectious diseases such as suspected sepsis, septic shock, urinary tract infection, acute appendicitis, and liver abscess[11-15]. Many recent studies have founded that the DNI was significantly associated with the severity of diseases related to sterile inflammation, such as out-of-hospital cardiac arrest, major traumatic injury, pulmonary embolism, acute myocardial infarction, and upper gastrointestinal hemorrhage[10,16-19]. As an inflammatory response to hypoperfusion and reperfusion owing to blood loss and tissue damage can develop into systemic inflammatory response syndrome, the DNI is a promising candidate for a biomarker for predicting the requirement of massive transfusion in patients with primary PPH in the ED[10,16]. In particular, the DNI can be useful in emergency settings because it can be rapidly and easily determined using an automated blood cell analyzer while measuring the complete blood count (CBC)[10,11,16].

7. Methods (page 7): How many patients were transferred from another hospital or a private clinic to ED? How many of them were transfused already before ED arrival? These baseline data should be presented in Table 1 or Figure 1.

Response: Thank you for your comment. This study enrolled only patients who were referred to our ED for evaluation and management of PPH from other hospitals or private obstetric clinics within the first 24 h of delivery. To determine the amount of transfusion, we measured both the amount of transfused blood before arrival at the ED and that after arrival at the ED (in the ED and intensive care unit [ICU] or general ward). We have added data for the patients who received blood transfusion before arrival at the ED in Table 1.

8. Methods (page 9): During the 10-yr study period, was the same instrument (ADVIA 2120) used continuously without change?

Response: Thank you for your comment. We have checked the information about the equipment. Our institution has consistently used the same equipment in the ED during the study period for research purposes. Replacement was made with an upgraded version of the same equipment (ADVIA 2120 -> ADVIA 2120i).

9. Methods (page 10): In line with my comment #7, clinical endpoints “the amount of transfused blood before ED arrival and the amount of transfused blood after ED arrival” can be presented in Table 1 or Figure 1.

Response: Thank you for your comment. Accordingly, we have added data on the amount of transfused blood before and after arrival at the ED in Table 1.

10. Statistical analysis: Although it is described that “The data are presented as medians and interquartile ranges for continuous variables”, the data are presented as mean/SD in Table 1. 

Response: Thank you for your comment. We have revised the manuscript accordingly, as follows:

“The data are presented as mean (SD) for continuous variables and as absolute or relative frequencies for categorical variables.”

11. Statistical analysis (page 11): diagnostic performance? diagnostic accuracy? Is it for the diagnosis of massive transfusion or for the prediction of massive transfusion? If it is for the prediction, throughout the text, predictive performance should be described instead of diagnostic performance.

Response: Thank you for your comment. We agree with your comment. Accordingly, we have revised “diagnostic” to “predictive” in the manuscript.

12. The whole “RESULTS” section has lots of duplicated descriptions of each Table and Figure. Please avoid duplicated presentation. Moreover, the first subsection, “Patients’ characteristics”, should be moved to the Methods section. In this retrospective study, this part is not the result corresponding to the study purpose but the basic description on the study population. 

Response: Thank you for your comment. We agree with you. Accordingly, we have moved the relevant text to the Methods section of the manuscript.

13. Table 2 can be modified to show the univariate analysis results together. That would be much more comprehensive.

Response: Thank you for your comment. We agree with your suggestion. We have revised Table 2 to show the univariate analysis results together.

14. Fig. 2: Did the authors compare AUC values among DNI, SI, DNI + SI? Please provide the statistical data (pairwise comparison of AUC values). The comparison of 0.821 vs. 0.831 (with overlapping CIs) does not seem to provide a significant result.

Response: Thank you for your comment. We agree with your comment. Accordingly, we have added pairwise comparison of the AUC values to supplement our results. We discussed this with our statisticians. We found that there was no significant difference between the SI and DNI+SI using pairwise comparison of the AUC values. However, the DNI+SI was significantly superior to the SI in the Net Reclassification Index (NRI) and Integrated Discrimination Index (IDI) analyses (see the response to reviewer’s comment no. 18). It is thought that this difference in results can be attributed to the AUC comparison, which has a more conservative tendency compared with the NRI or IDI. 

15. Discussion section is quite lengthy spanning 4.5 pages (including Conclusion). It should be shortened compactly. The third paragraph (pages 14 – 16) is full of general descriptions that are not directly related to the study results. In the fourth paragraph (page 16), what was the lactate levels in this study population? If lactate levels are so important (as described in this paragraph), it deserves to be included in Table 1 and can be analyzed together.

Response: Thank you for your comment. We agree with your comment. We have reduced the Discussion section accordingly, as shown below. In addition, we have added the lactate levels in Table 1.

“The DNI is available in emergency settings because it can be rapidly and easily determined using an automated blood cell analyzer while measuring the CBC[10,11,16] The DNI value is correlated with the manual count of immature granulocytes[11]. However, automatic calculation of the DNI is a more prompt and accurate method than manual counting of immature granulocytes[10]. The diagnostic criteria for systemic inflammatory response syndrome include an increase in the number of immature granulocytes in the circulation as an important factor[10,16]. Several studies have shown that an increase in the DNI can predict the severity of infection and sterile inflammation without infection and that it is significantly associated with poor outcomes in specific disease conditions[10,13-18]. Massive hemorrhage causes detrimental systemic inflammation without infection[10,16]. The sustained, exacerbated inflammatory response is strongly associated with increased mortality[10,16]. Emergency granulopoiesis is the most accepted theory for explaining why an elevated DNI value is associated with the use of massive transfusion after primary PPH[21,22]. Emergency granulopoiesis implies the expansion of the immature granulocytic population for providing hematopoietic precursors to sustain the increasing number of circulating neutrophils[10,21,22]. The number of circulating immature neutrophils can be rapidly increased to compensate for the destruction of mature neutrophils due to severe systemic inflammation and the loss of active neutrophils due to massive consumption related to sterile stimuli, such as chemical agents, trauma, bleeding, and ischemic insult[21,23,24]. Livingston et al. proposed that bone marrow failure can develop due to insufficient perfusion and ischemia during hemorrhagic shock[25,26]. This bone marrow dysfunction causes a transient failure of granulopoiesis[25,26]. Consequently, severe hemorrhage may contribute to the release of immature granulocytes into the circulation[23,27]. As pregnant women often have prenatal anemia, it is difficult to determine the occurrence of massive hemorrhage and evaluate the requirement for massive transfusion based on hemoglobin levels alone in patients with PPH[28]. Although lower hemoglobin or hematocrit levels have been widely and interchangeably used as indicators of severe bleeding, their diagnostic values for determining the severity of blood loss in the initial phase in patients with severe hemorrhage, such as trauma-related and GI bleeding, remain controversial[28]. In the present study, the hemoglobin level on ED admission was not associated with the requirement for massive transfusion. 

In addition, some studies have reported that the initial SI value, as a hemodynamic indicator based on the HR and SBP (easily obtained in the ED), could predict the requirement for massive transfusion. Although the initial SI value on ED admission was independently associated with the requirement for massive transfusion in patients with primary PPH, it had a relatively low specificity[7,8]. Sohn et al. revealed that combining the initial SI value with the lactate level could improve its predictive performance for the requirement for massive transfusion in patients with primary PPH[7]. As increased plasma lactate levels are significantly associated with increased morbidity and mortality, lactate measurement is a valuable tool for triage and risk stratification. Hyperlactatemia is often attributed to tissue hypoxia in several diseases[29]. Conversely, an increased plasma lactate level might not always signify hypoxia, thus reducing its specificity[30]. Hyperlactatemia without hypoxia might result from aerobic glycolysis during sepsis, liver failure, altered pyruvate dehydrogenase activity, hyperventilation, and elevated catecholamine levels; it might also occur in the presence of certain drugs and toxins[31-33]. This may negatively affect the usefulness of lactate for predicting disease severity.”

16. This study used only the initial DNI values at ED admission. Follow-up DNI values would be available in these patients. Considering the time definition of massive transfusion, what was the DNI value and shock index value after 24 hr?

Response: Thank you for your helpful comment. In line with your comment, we have analyzed the DNI and SI values at 24 h after ED admission. In these results, the DNI value was significantly higher in patients with requirement for MT than in those with no requirement for MT. However, there was no significant difference in the SI at 24 h after ED admission between the two groups.

17. In Table 1, how many patients had shock index > 1.0 in the two groups? The laboratory data should be presented in SI units.

Response: Thank you for your comment. We have revised Table 1 accordingly. Among all patients, 94 (34%) had an SI >1.0. There were significantly more patients with an SI >1.0 among those with requirement for MT than among those with no requirement for MT.

18. In addition to the ROC curve analysis, IDI/NRI analyses are recommended to show additive value (any superiority) of DNI value to shock index.

Response: Thank you for your comment. We agree with your comment. Accordingly, we have added pairwise comparison of the AUC values to supplement our findings. We found that there was no significant difference between the SI and DNI+SI using pairwise comparison of the AUC values. However, the DNI+SI was significantly superior to the SI in the NRI and IDI analyses. It is thought that this difference in results is caused by the AUC comparison, which has a more conservative tendency compared with the NRI or IDI.

Overall, the readability and organization of this manuscript is acceptable. 

However, the additive value of DNI value on top of shock index should be more emphasized to support their conclusion. Although the authors mentioned “screening tool” and “additional prediction tool” as the clinical usefulness of DNI value on top of shock index (page 17), which change in detail can we expect in the clinical practice? Can we decrease the transfused blood amount? Can we decrease the number of patients who were transfused massively? The authors performed a retrospective analysis; so, if they want to emphasize the adding value of DNI onto shock index, they can propose an algorithm (including both DNI value and shock index) for the clinical decision-making and can show the simulation results how the massive transfusion practice would have been changed.

Response: Thank you for your helpful comment. We have discussed this with our statisticians. In this study, we demonstrated that the DNI was independently associated with the requirement for MT in patients with PPH. In comparison with the SI alone, the addition of the DNI to the SI showed significant increases in specificity, accuracy, LR+, and post-test probability if the test was positive. However, there was no improvement in sensitivity, LR-, and post-test probability if the test was negative. Nevertheless, we believe that the application of the DNI has its own clinical value in improving the prediction of hemorrhage severity and requirement for MT in patients with PPH.

In this study, MT was defined as a transfusion of ≥ 10 units of packed red blood cells. MT is not a simple medical practice that is decided based on the initial condition of patients in the ED; it requires monitoring of laboratory results and vital signs and medical judgment based on the changes of patient’s condition during the critical period. 

As resources like blood tests and ICU are limited in the ED, it is important to indirectly estimate the severity of the hemorrhage, the probability of MT requirement, or the actual amount of bleeding. Considering the limited medical resources, it is of great importance to improve the specificity of prediction for the requirement for MT in the ED. 

This study enrolled only patients who were referred to our ED for evaluation and management of PPH from other hospitals or private obstetric clinics within the first 24 h of delivery. Basically, we applied standardized protocols (SPEED CP) to these patients and evaluated patients with PPH regarding requirement for high-level monitoring and MT. For these patients, we believe that it is clinically meaningful to screen for the need for MT accurately using the DNI value obtained through a simple CBC test. It is difficult to clearly determine the requirement for MT in patients with PPH using only the DNI and DNI+SI values; however, we believe that these would be of great clinical utility as auxiliary indicators in patients with PPH. Of course, it is unfortunate that the sensitivity, LR-, and post-test probability in cases of negative tests did not improve after adding the DNI to the SI. Further study is needed to clarify the advantages and disadvantages. 

Regarding the algorithm suggested by the reviewer, the DNI+SI can clearly increase the likelihood ratio for requirement for MT. However, clinical decision making for MT requirement is a very complex process that combines assessments drawn by monitoring of laboratory results and vital signs and medical judgment based on the changes of patient’s condition over a considerable period of time. Therefore, we think that establishing an algorithm related to MT requirement determination using only this value may confuse readers. Rather than determining the requirement for MT in patients with PPH based on the DNI and DNI+SI values alone, it would be appropriate to use them as auxiliary indicators. Based on your comment, we have added this point in the manuscript, as follows: 

“Although clinical decision making for MT requirement is a very complex process in the ED, further study is needed to establish a clinical algorithm for determining the requirement for MT in patients with PPH based on the DNI and DNI+SI values alone.”

Reviewer #2: 

The authors demonstrate a complex statistical association to integrate into daily medical practice. Moreover, the primary and secondary objectives are not at all clear. The method lacks precision; it is a study with a change of initial destination, but what was the purpose of the first study? what were the criteria? The sample is small and few values are really significant. This article does not provide any real benefit to the medical literature

Response: Thank you for your helpful comment. We understand that you have raised questions about the purpose and clinical applicability of this study. Considering the fact that there are not enough useful predictors of hemorrhage severity in patients with PPH, this study investigated the possibility of predicting the requirement for MT using the DNI, which has recently been found useful in predicting the severity of sterile inflammatory disease (out of hospital cardiac arrest, trauma, ST elevation MI, and upper GI bleeding). 

In this study, we demonstrated that the DNI was independently associated with the requirement for massive transfusion in patients with PPH. In comparison with the SI alone, the addition of the DNI to the SI showed significant increases in the specificity, accuracy, LR+, and post-test probability if the test was positive. However, there was no improvement in sensitivity, LR-, and post-test probability if the test was negative. Nevertheless, we believe that the application of the DNI has its own clinical value in improving the prediction of hemorrhage severity and requirement for MT in patients with PPH.

In this study, MT was defined as a transfusion of ≥ 10 units of packed red blood cells. MT is not a simple medical practice that is decided based on the initial condition of patients in the ED, but it is a medical practice that combines monitoring of laboratory results and vital signs and medical judgment based on the changes of patient’s condition during the critical period. 

As resources like blood tests and ICU are limited in the ED, it is important to indirectly estimate the severity of hemorrhage, the probability of MT requirement, or the actual amount of bleeding. Considering the limited medical resources, it is of great importance to improve the specificity of MT prediction in the ED. 

This study enrolled only patients who were referred to our ED for evaluation and management of PPH from other hospitals or private obstetric clinics within the first 24 h of delivery. Basically, we applied standardized protocols (SPEED CP) to these patients and evaluated patients with PPH regarding requirement for high-level monitoring and MT. For these patients, we believe that it is clinically meaningful to screen for the need for MT accurately using the DNI value obtained through a simple CBC test. It is difficult to clearly determine the requirement for MT in patients with PPH using only the DNI and DNI+SI values; however, we believe that these would be of great clinical utility as auxiliary indicators in patients with PPH. 

Reviewer #3: 

Dear Editor,

Thanks for the opportunity to read the article.

It was a well-written and well-designed study that should be accepted for publication.

My recommendation is to accept the article in its current form.

Best regards.

Response: We greatly appreciate your comments. 

Reviewer #4: 

The manuscript lines are not numbered, so I identified the lines of the manuscript by quoting parts of them.

Response: We apologize for the inconvenience. We have revised the resubmitted manuscript accordingly.

Introduction

“…is the cause of maternal mortality in 12% of cases…” I assume what you mean is that 12% of maternal mortality is caused by PPH. Is that correct?

Response: Thank you for pointing this out. We apologize for the confusion. We have revised the text accordingly, as follows:

“PPH constitutes a major risk for maternal mortality and morbidity in both developed and developing countries and is the cause of maternal mortality in 12% of cases in the United States.

PPH constitutes a major risk for maternal mortality and morbidity in both developed and developing countries and causes 12% of maternal mortality in the United States.”

“An SI of >1.0 showed a specificity of 78.7%, a sensitivity of 59.2%, a positive predictive value (PPV) of 58.7%, and a negative predictive value (NPV) of 79.1%.” Predictive values, although often reported, are useless statistics because they change with the prevalence of disease and you are trying to estimate the prevalence of disease in the individual patient. predictive values have almost no clinical usefulness, so please remove them from the paper. 

Response: Thank you for your helpful comment. We agree with your comment. We have deleted the statistics on PPV and NPV in the revised manuscript and Tables and added likelihood ratios in our results.

Although sensitivities and specificities are somewhat useful, likelihood ratios are more useful since they can be applied to the individual patient. Therefore, better would be a statement such as, “Using the results from reference 7, for a SI > 1.0, the positive and negative likelihood ratios for massive transfusion in patients with PPH are 3 and the negative likelihood ratio is 0.5, making SI of little help to the clinician for predicting the need for transfusion.” In general likelihood ratios are useful only if they are >5 or <0.2.

Response: Thank you for your helpful comment. We agree with your comment. We have revised the manuscript accordingly, as follows:

“An SI of >1.0 showed a positive likelihood ratio of 2.8 and a negative likelihood ratio of 0.5 for the requirement for massive transfusion in patients with PPH, indicating that the SI is of little help to clinicians for predicting the need for transfusion [7]. In general, likelihood ratios are only useful if their values are >5 or <0.2.”

Methods

“…and identified…PPV, and NPV…” As mentioned above, PPV and NPV are useless statistics. Please eliminate them from the paper.

Response: Thank you for your helpful comment. As per your suggestion, we have deleted the statistics on PPV and NPV from the revised manuscript.

Results

“…SBP (94.267…). Reporting SBP to 3 significant figures is not needed. Better to report the results for blood pressures and heart rates as whole numbers. This will also make the paper easier to read.

Response: Thank you for your helpful comment. Accordingly, we have revised the systolic blood pressure and heart rate values as whole numbers in the revised manuscript.

I request two general changes in the paper. One has to do with the presentation of the results. As mentioned above, please remove predictive values and include likelihood ratios in your results. 

Response: Thank you for your helpful comment. We have deleted the statistics on PPV and NPV in the revised manuscript and Tables and added likelihood ratios in our results.

The second problem is the interpretation of the ROC curves as described next.

Fig. 2 The ROC curves for the DNI and DNI+SI are almost overlapping so it is slightly surprising that your sensitivities and specificities (Table 3) are so different. I think this is most likely because you have chosen different points on the curves to calculate them. 

I have described the method I use below, that I think is preferable. Nonetheless, the likelihood ratios calculated from the sensitivities and specificities listed in Table 3 are shown in the following table. Using these likelihood ratios, I have also calculated the post-test probabilities for need for massive blood transfusion, assuming that the pretest probability is 21.6% (60/278).

LR+ LR- Post-test probability if test positive Post-test probability if test is negative

DNI 3.5 0.48 49% 11%

SI 3.3 0.31 47% 8%

DNI + SI 11.6 0.49 76% 12%

The pretest probability of MT requirement is 21.6%. The question is, does applying DNI and/or SI change that significantly enough to benefit the clinician seeing the next patient that may require MT (and, if so, what does the clinician do differently than would not have been done if this information was not available)? As you can see, adding DNI to SI is not helpful if the test is negative, but somewhat helpful if the test is positive. In general, applying the results of your study to patients with PPH would reduce the probability of requiring MT in patients with a negative test by half, yet 10% of those patients still require MT. On the other hand, of those with a positive test, about one quarter will end up not requiring MT, using the DNI+SI parameters. Please comment on how this would help the clinician actually caring for these patients, in particular, once this result is obtained, how does the plan of the physician change? Would the clinician then immediately initiate the massive transfusion protocol, knowing that about one-quarter of the patients do not need it? I think that this type of analysis points out that even when differences are highly statistically significant, this does not necessary translate into a result that is that useful to the clinician seeing the next patient with PPH.

I performed two other calculations on your data. First, I had some difficulty with data taken from the DNI+SI ROC graph, since the point on the graph where sensitivity = 53.5% and specificity = 95.4% is not actually on the line. However, the point on the SI and SI+DNI lines closest to this point show the following results:

LR+ LR- Post-test probability if test positive Post-test probability if test is negative

SI 5.6 0.54 60% 13%

DNI + SI 7.3 0.53 67% 13%

This suggests that adding DNI to the SI is only marginally helpful. This is particularly important for two reasons: 1. The SI is immediately available once vital signs are taken and 2. Many clinicians may not be familiar with DNI and/or may have difficulty remembering the formula to calculate it or be able to calculate it themselves.

I actually calculated the same values by finding the points on the curves in Fig. 2 that are closest to the left upper corner of the graph (where sensitivity and specificity = 1). I did this by printing the graph and using a compass to find the point. This produced the following table, corresponding to the one above.

LR+ LR- Post-test probability if test positive Post-test probability if test is negative

DNI 2.8 0.45 43% 11%

SI 3.8 0.30 41% 7%

DNI+SI 5.0 0.35 57% 9%

As you can see, the results are similar to yours except for the DNI+SI with a positive result. This also suggests that adding DNI to SI is only a little helpful If the test is positive, but not at all if the test is negative.

Response: We are extremely grateful for the time you have taken to review our data. In this study, we demonstrated that the DNI was independently associated with the requirement for massive transfusion in patients with PPH. In comparison with the SI alone, the addition of the DNI to the SI showed significant increases in specificity, accuracy, LR+, and post-test probability if the test was positive. However, there was no improvement in sensitivity, LR-, and post-test probability if the test was negative. Nevertheless, we believe that the application of the DNI has its own clinical value to improve the prediction of hemorrhage severity and requirement for massive transfusion in patients with PPH.

In this study, MT was defined as a transfusion of ≥ 10 units of packed red blood cells. MT is not a simple medical practice that is decided based on the initial condition of patients in the ED, but it is a medical practice that combines monitoring of laboratory results and vital signs and medical judgment based on the changes of patient’s condition over a considerable period of time. 

As resources like blood tests and ICU are limited in the ED, it is important to indirectly estimate the severity of hemorrhage, the probability of MT requirement, or the actual amount of bleeding. Considering the limited medical resources, it is of great importance to improve the specificity of prediction of the need for MT. 

This study enrolled only patients who were referred to our ED for evaluation and management of PPH from other hospitals or private obstetric clinics within the first 24 h of delivery. Basically, we applied standardized protocols (SPEED CP) to these patients and evaluated patients with PPH regarding requirement for high-level monitoring and MT. For these patients, we believe that it is clinically meaningful to screen for the need for MT accurately using the DNI value obtained through a simple CBC test. It is difficult to clearly determine the requirement for MT in patients with PPH using only the DNI and DNI+SI values; however, we believe that these would be of great clinical utility as auxiliary indicators in patients with PPH. Of course, it is unfortunate that the sensitivity, LR-, and post-test probability in cases of negative tests did not improve after adding the DNI to the SI. Further study is needed to clarify the advantages and disadvantages of this approach. Rather than determining the requirement for MT in patients with PPH based on the DNI and DNI+SI values alone, it would be appropriate to use them as auxiliary indicators.

---

## [Decision Letter · Decision Letter 1]

9 Aug 2021

PONE-D-20-31096R1

Delta neutrophil index and shock index can predict the requirement for massive transfusion in patients with primary postpartum hemorrhage in the emergency department

PLOS ONE

Dear Dr. You,

Thank you for submitting your manuscript to PLOS ONE. After careful consideration, we feel that it has merit but does not fully meet PLOS ONE’s publication criteria as it currently stands. Therefore, we invite you to submit a revised version of the manuscript that addresses the points raised during the review process.

ACADEMIC EDITOR: Our expert reviewers still have raised a number of points which we believe major modifications are necessary to improve the revised manuscript, taking into account the reviewers' remarks below.

We look forward to receiving your revised manuscript.

Kind regards,

Wisit Cheungpasitporn, MD

Academic Editor

PLOS ONE

Reviewers' comments:

Reviewer's Responses to Questions

**Comments to the Author**

1. If the authors have adequately addressed your comments raised in a previous round of review and you feel that this manuscript is now acceptable for publication, you may indicate that here to bypass the “Comments to the Author” section, enter your conflict of interest statement in the “Confidential to Editor” section, and submit your "Accept" recommendation.

Reviewer #4: (No Response)

2. Is the manuscript technically sound, and do the data support the conclusions?

Reviewer #4: No

3. Has the statistical analysis been performed appropriately and rigorously? 

Reviewer #4: No

4. Have the authors made all data underlying the findings in their manuscript fully available?

Reviewer #4: Yes

5. Is the manuscript presented in an intelligible fashion and written in standard English?

Reviewer #4: Yes

6. Review Comments to the Author

Reviewer #4: Title: “Delta neutrophil index and shock index can predict the requirement…” I think the use of the word “predict” offers something to the readers that the study does not actually provide. Perhaps better would be, “Delta neutrophil index and shock index can risk stratify the requirement…”

Although the information in the “Abstract” is correct, it is of little interest to the clinician. That the odds ratios are statistically significant is of little help to the clinician about to care for the next patient with PPH. It would be better to substitute information about likelihood ratios since they can be used directly by the clinician to take care of the patient. Even if the authors choose to retain odds ratios in the abstract, they should also include the information that values below the cutoff were not helpful, whether they were statistically significant or not.

I had previously made the comment about the number of significant figures: “…SBP (94.267…). Reporting SBP to 3 significant figures is not needed. Better to report the results for blood pressures and heart rates as whole numbers. This will also make the paper easier to read.

The response was: “Thank you for your helpful comment. Accordingly, we have revised the systolic blood pressure and heart rate values as whole numbers in the revised manuscript.” However, I did not see where this was actually done. Please follow up with this. Furthermore, Table 1 would be easier to read if values were rounded out to whole numbers. This would not lose anything in terms of conclusions reached by the study. For example, the first entry, 32.70±3.95 is better reported as 33 ± 4. There are other places in the body of the paper where results are given with an inordinately large number of significant figures. This just makes the paper harder to read, does not add anything to the interpretation of the results and is actually not justified from a statistical point-of-view. For example, “…with a specificity of 83.03% (95% CI: 78.05–88.01)” implies that the specificity of 83.04 is incorrect, whereas the 95% CI includes that value. I would be better to say, “…with a specificity of 83% (95% CI: 78-88).” Furthermore, I really don’t think anyone cares whether the specificity is 83.03% or 83%.

Regarding the following sentence: “On combining a DNI value of ≥3.3 with an SI value of ≥1.0, the specificity and PLR increased to 95.41% (95% CI; 92.63–98.19) and 11.63 (95% CI: 6.07–22.27), respectively.” I had some difficulty with data taken from the DNI+SI ROC graph, since the point on the graph where sensitivity = 53.5% and specificity = 95.4% is not actually on the line (see graph in attachment, point A). The authors did not address this concern in their response to the initial review even though this is crucial for interpreting the data. Their result (PLR for DNI+SI = 11.63) is much higher than the 7.3 or 5.0 that I calculated using points that are actually on the plotted line (attached graph, points B and C). The authors need to address this since you cannot arbitrarily choose any point on the graph to make the calculations. You need to use a point that is actually on the line in the graph. Please make this correction in the manuscript.

7. PLOS authors have the option to publish the peer review history of their article (what does this mean?). If published, this will include your full peer review and any attached files.

Reviewer #4: No

---

## [Author Response · Author response to Decision Letter 1]

29 Sep 2021

Please, check additional file for review resposne to reviewer's comments.

*** Response to reviewer’ s comments ***

Delta neutrophil index and shock index can stratify risk for the requirement for massive transfusion in patients with primary postpartum hemorrhage in the emergency department

Response to reviewer #4’s comments

Reviewer #4: Title: “Delta neutrophil index and shock index can predict the requirement…” I think the use of the word “predict” offers something to the readers that the study does not actually provide. Perhaps better would be, “Delta neutrophil index and shock index can risk stratify the requirement…” Although the information in the “Abstract” is correct, it is of little interest to the clinician. That the odds ratios are statistically significant is of little help to the clinician about to care for the next patient with PPH. It would be better to substitute information about likelihood ratios since they can be used directly by the clinician to take care of the patient. Even if the authors choose to retain odds ratios in the abstract, they should also include the information that values below the cutoff were not helpful, whether they were statistically significant or not. 

Response: Thank you for your astute comments and valuable suggestions. We agree with the reviewer’s opinion.

First, we have revised the title. 

Delta neutrophil index and shock index can stratify risk for the requirement for massive transfusion in patients with primary postpartum hemorrhage in the emergency department

Second, we have substituted odds ratios with likelihood ratios.

Results: Multivariable logistic regression revealed that the DNI and SI were independent predictors of MT. The optimal cut-off values of ≥3.3% and ≥1.0 for the DNI and SI, respectively, were significantly associated with an increased risk of MT (DNI: positive likelihood ratio [PLR] 3.54, 95% confidence interval [CI] 2.5–5.1 and negative likelihood ratio [NLR] 0.48, 95% CI 0.4–0.7; SI: PLR 3.21, 95% CI 2.4–4.2 and NLR 0.31, 95% CI 0.19–0.49). The optimal cut-off point for predicted probability was calculated for combining the DNI value and SI value with the equation derived from logistic regression analysis. Compared with DNI or SI alone, the combination of DNI and SI significantly improved the specificity, accuracy, and positive likelihood ratio of the MT risk. 

I had previously made the comment about the number of significant figures: “…SBP (94.267…). Reporting SBP to 3 significant figures is not needed. Better to report the results for blood pressures and heart rates as whole numbers. This will also make the paper easier to read. 

The response was: “Thank you for your helpful comment. Accordingly, we have revised the systolic blood pressure and heart rate values as whole numbers in the revised manuscript.” However, I did not see where this was actually done. Please follow up with this. Furthermore, Table 1 would be easier to read if values were rounded out to whole numbers. This would not lose anything in terms of conclusions reached by the study. For example, the first entry, 32.70±3.95 is better reported as 33 ± 4. There are other places in the body of the paper where results are given with an inordinately large number of significant figures. This just makes the paper harder to read, does not add anything to the interpretation of the results and is actually not justified from a statistical point-of-view. For example, “…with a specificity of 83.03% (95% CI: 78.05–88.01)” implies that the specificity of 83.04 is incorrect, whereas the 95% CI includes that value. I would be better to say, “…with a specificity of 83% (95% CI: 78-88).” Furthermore, I really don’t think anyone cares whether the specificity is 83.03% or 83%.

Response: Thank you for your pertinent comments. We sincerely apologize for our oversight in the previous manuscript version. We have revised the values to whole numbers in this manuscript version.

Regarding the following sentence: “On combining a DNI value of ≥3.3 with an SI value of ≥1.0, the specificity and PLR increased to 95.41% (95% CI; 92.63–98.19) and 11.63 (95% CI: 6.07–22.27), respectively.” I had some difficulty with data taken from the DNI+SI ROC graph, since the point on the graph where sensitivity = 53.5% and specificity = 95.4% is not actually on the line (see graph in attachment, point A). The authors did not address this concern in their response to the initial review even though this is crucial for interpreting the data. Their result (PLR for DNI+SI = 11.63) is much higher than the 7.3 or 5.0 that I calculated using points that are actually on the plotted line (attached graph, points B and C). The authors need to address this since you cannot arbitrarily choose any point on the graph to make the calculations. You need to use a point that is actually on the line in the graph. Please make this correction in the manuscript.

Response: Thank you for your remarkable comments and insightful suggestions. As you mentioned, on combining a DNI value of ≥ 3.3 with an SI value of ≥ 1.0, an optimal cut-off point could not be represented using ROC curves. Therefore, we determined arbitrary point as an optimal cut-off point by combining the optimal cut-off points of both DNI value of ≥ 3.3 and SI value of ≥ 1.0. We believe that this optimal cut-off point can help in the clinical decision-making. However, we agree that we should measure sensitivity, specificity, and likelihood ratio based on the optimal cut-off point obtained on combining the DNI value and SI value represented on the ROC curve. We discussed this point with our statisticians and added the optimal cut-off point to the ROC curve using the equation derived from logistic regression. 

To stratify risk for the requirement for massive transfusion by combining the DNI value and SI value in patients with primary postpartum hemorrhage in the ED, we used the following equation derived from logistic regression. 

Predicted probability = 1/(1+exp (5.348-0.2464*DNI-3.2691.*SI)

Based on this equation, we calculated predicted probability for combining the DNI value and SI value. We determined the optimal cut-off point for predicted probability (DNI+SI) to be ≥ 0.249.

When we stratified risk for the requirement for massive transfusion with the predicted probability (≥ 0.249) by combining the DNI value and SI value in patients with primary postpartum hemorrhage in the ED, the application of the value significantly improved the specificity, sensitivity, and PLR for massive transfusion in comparison with SI alone. In addition, it also improved PLR and NLR in comparison with DNI alone. We have marked a “cross” on the ROC curve of Figure 2. This point represents the optimal cut-off point based on the predicted probability (DNI+SI) of ≥ 0.249.

We have deleted texts pertaining to the optimal cut-off point derived by combining the optimal cut-off points of both DNI value of ≥3.3 and SI value of ≥1.0 from the manuscript according to reviewer’s comment. We have added the optimal cut-off point to the ROC curve using the equation derived from logistic regression to the revised manuscript.

Method

In addition, to stratify risk for the requirement for massive transfusion by combining the DNI value and SI value, we used the equation derived from logistic regression analysis. Based on this equation, we calculated the predicted probability for combining the DNI value and SI value. We determined optimal the cut-off point for the predicted probability. Based on this optimal cut-off point, we measured sensitivity, specificity, PLR, NLR, and accuracy.

Results

The predicted probability derived from the logistic equation was computed for each patient. When the predicted probability of the equation [Predicted probability = 1/(1+exp(5.348-0.2464*DNI-3.2691.*SI)] containing the combination of the DNI and SI was more than 0.249, the specificity was 87% (95% CI: 82–91), sensitivity was 70% (95% CI: 58–82), accuracy was 83% (95% CI: 79–88), PLR was 5.26 (95% CI: 3.6–7.7), and NLR was 0.35 (95% CI: 0.2–0.5) for MT. The combination of the DNI and SI significantly improved the specificity (DNI vs DNI + S, p<0.001; SI ≥1.1 vs DNI + SI, p<0.001), accuracy (DNI vs DNI + SI, p<0.001; SI ≥1.1 vs DNI + SI, p<0.001), and PLR (DNI vs DNI + S, p<0.001; SI ≥1.1 vs DNI + SI, p=0.038; using the GEE method) for the requirement for massive transfusion when compared with either DNI or SI (Table 3).

---

## [Decision Letter · Decision Letter 2]

4 Oct 2021

Delta neutrophil index and shock index can stratify risk for the requirement for massive transfusion in patients with primary postpartum hemorrhage in the emergency department

PONE-D-20-31096R2

Dear Dr. You,

We’re pleased to inform you that your manuscript has been judged scientifically suitable for publication and will be formally accepted for publication once it meets all outstanding technical requirements.

Kind regards,

Wisit Cheungpasitporn, MD

Academic Editor

PLOS ONE

Additional Editor Comments (optional):

I reviewed the revised manuscript and the response to reviewers' comments. Revised Manuscript is well written. All comments have been addressed and thus accepted for publication.

Reviewers' comments:

Reviewer's Responses to Questions

**Comments to the Author**

1. If the authors have adequately addressed your comments raised in a previous round of review and you feel that this manuscript is now acceptable for publication, you may indicate that here to bypass the “Comments to the Author” section, enter your conflict of interest statement in the “Confidential to Editor” section, and submit your "Accept" recommendation.

Reviewer #4: (No Response)

2. Is the manuscript technically sound, and do the data support the conclusions?

Reviewer #4: Yes

3. Has the statistical analysis been performed appropriately and rigorously? 

Reviewer #4: Yes

4. Have the authors made all data underlying the findings in their manuscript fully available?

Reviewer #4: Yes

5. Is the manuscript presented in an intelligible fashion and written in standard English?

Reviewer #4: Yes

6. Review Comments to the Author

Reviewer #4: (No Response)

7. PLOS authors have the option to publish the peer review history of their article (what does this mean?). If published, this will include your full peer review and any attached files.

Reviewer #4: No

---

## [Editor Report · Acceptance letter]

8 Oct 2021

PONE-D-20-31096R2 

Delta neutrophil index and shock index can stratify risk for the requirement for massive transfusion in patients with primary postpartum hemorrhage in the emergency department 

Dear Dr. You:

I'm pleased to inform you that your manuscript has been deemed suitable for publication in PLOS ONE. Congratulations! Your manuscript is now with our production department. 

Kind regards, 

on behalf of

Dr. Wisit Cheungpasitporn 

Academic Editor

PLOS ONE